

# Associations between meteorological factors and pregnancy complications during different pregnancy trimesters: a multicenter retrospective study in eastern China

Tao Chen[1,*], Jie Cai[2,*], Da He[3], Hui Zhu[4], Xiang Chen[3], Yuqiong Li[1], Wen Ye[1], Bingqi Li[1], Qinyan Xu[5], Lindan Ji[6,7] and Jin Xu[1,6]

[1] School of Public Health, Health Science Center, Ningbo University, Ningbo, Zhejiang, China
[2] Center for Reproductive Medicine, Women and Children's Hospital of Ningbo University, Ningbo, Zhejiang, China
[3] Department of Obstetrics and Gynecology, Yinzhou District Maternal and Child Health Care Institute, Ningbo, Zhejiang, China
[4] Department of Internal Medicine, Health Science Center, Ningbo University, Ningbo, Zhejiang, China
[5] Department of Obstetrics and Gynecology, The 906th Hospital of PLA, Ningbo, Zhejiang, China
[6] Zhejiang Key Laboratory of Pathophysiology, Health Science Center, Ningbo University, Ningbo, Zhejiang, China
[7] Department of Biochemistry and Molecular Biology, School of Basic Medical Sciences, Health Science Center, Ningbo University, Ningbo, Zhejiang, China
[*] These authors contributed equally to this work.

Corresponding authors
Lindan Ji, jilindan@nbu.edu.cn
Jin Xu, xujin1@nbu.edu.cn

## ABSTRACT

**Background**. Previous studies have demonstrated the effects of ambient temperature on the risk of pregnancy complications. However, the associations between multiple meteorological factors and pregnancy complications have rarely been studied.

**Methods**. We carried out a retrospective study on the impacts of meteorological factors on pregnancy complications in different trimesters in Ningbo, China, from 2013–2023. Daily meteorological factors data were obtained from the National Aeronautics and Space Administration (NASA). Moreover, a meteorological factor score (MFS) was calculated. Logistic regression models were applied to assess the effects of individual meteorological factors and MFS on pregnancy complications during different trimesters. Distributed lag nonlinear models were used to explore the sensitive time windows of extreme meteorological factors in different weeks of gestation. The interaction effects of extreme meteorological factors were assessed.

**Results**. A total of 92,332 participants were included in the study, with diagnoses as follows: gestational diabetes mellitus (GDM) in 17,814 participants (19.29%), gestational hypertension (GH) in 3,860 participants (4.18%), preeclampsia (PE) in 3,101 participants (3.36%), and hypothyroidism in 17,418 participants (18.86%). Participants in the highest MFS quintile during the first trimester had greater risks of GDM (aOR = 1.186, 95% CI [1.079–1.304]), GH (aOR = 1.596, 95% CI [1.323–1.925]), PE (aOR = 1.347, 95% CI [1.094–1.658]) and hypothyroidism (aOR = 1.257, 95% CI [1.141–1.385]) compared to the lowest quintile. Extreme meteorological exposures influenced complications within distinct windows: risks of GDM, GH, and PE concentrated in mid-pregnancy (3rd–5th months), while hypothyroidism showed

first-trimester vulnerability. Interactions between meteorological factors collectively influenced the risk of hypothyroidism.

**Conclusions**. Our findings demonstrated that elevated temperature, increased precipitation, prolonged sunshine duration, and reduced surface pressure were critical risk factors for pregnancy complications. Targeted protective measures should be taken to reduce the risk of pregnancy complications.

# INTRODUCTION

In recent decades, global meteorological factors have changed dramatically and have been correlated with a threat to global health (*Costello et al., 2009*; *Romanello et al., 2023*). These environmental determinants, encompassing key parameters such as temperature, humidity, air pressure, precipitation, wind speed, and sunshine duration, collectively exacerbate health risks for the population through multifaceted pathways—from metabolic dysregulation to cardiorespiratory impairments and reproductive complications (*Chen et al., 2018*; *Andhikaputra et al., 2023*; *Ratter-Rieck, Roden & Herder, 2023*). Climate change is driving the global increase in the frequency, intensity and duration of extreme temperatures. Ambient high temperatures are associated with heat-related morbidity, cardiac workload augmentation, and exacerbation of cardiovascular symptoms, whereas cold exposure correlates with hypothermia risk and increased myocardial ischemia incidence (*Faurie et al., 2022*). Concurrently, climate change exacerbates air pollution by elevating concentrations of particulate matter and ozone, adversely impacting respiratory, cardiovascular, and reproductive health (*Segal & Giudice, 2022*; *Liang et al., 2023*). Additionally, these climate-driven environmental stressors—including temperature variability, air pollutants, ultraviolet radiation, and psychological stress—disrupt hormonal balance, immune response, and endocrine regulation through interconnected physiological pathways (*Yüzen et al., 2023*; *Hannan et al., 2024*).

Emerging epidemiological evidence demonstrated that meteorological factors also affected the health of pregnant women and exhibited trimester-specific associations with pregnancy complications, such as gestational diabetes mellitus (GDM), gestational hypertension (GH) and preeclampsia (PE) (*Yan et al., 2022*). Toronto study data revealed a dose–response relationship between pre-screening outdoor temperature exposure (each 10 °C increase over 30 days) and GDM risk (*Booth et al., 2017*). While Guangzhou research identified U-shaped risk curves linking both extremely high and low temperatures during the second trimester with GDM susceptibility, and discovered a positive correlation between diurnal temperature range (DTR) and GDM risk (*Zhang et al., 2021*). A systematic review demonstrated different thermal effects: heat exposure increased the risk of PE and GH, whereas cold exposure decreased the risk (*Mao et al., 2023*). Seasonality also associated with

the incidence of GDM and hypertensive disorders of pregnancy (HDP), while multicenter evidence implicated solar radiation in hypertension pathogenesis (*Chiefari et al., 2017*; *Rohr Thomsen et al., 2020*; *Lu et al., 2022*).

While substantial evidence has established extreme temperature as a critical predictor of pregnancy complications, significant knowledge gaps persist regarding other meteorological determinants. Current literature predominantly focuses on thermal extremes, with limited characterization of humidity, surface pressure, wind speed, precipitation, and sunshine duration impacts (*Lakhoo et al., 2025*; *Preston et al., 2020*). More critically, the field lacks of comprehensive evaluations of cumulative meteorological exposures and their interactive effects on the risk of pregnancy complications.

This study was based on a multicenter retrospective study in eastern China that included 92,332 people. In this study, we aimed to comprehensively evaluate the independent risk effects of different meteorological factors on pregnancy complications and construct a comprehensive meteorological factor score (MFS) to analyze the correlation between the overall exposure level of meteorological factors and disease. These results could play a positive role in the early prevention of pregnancy complications in a time of increasing climate change.

## MATERIALS & METHODS

### Study design and participants

This study is based on a multicenter retrospective cross-sectional study conducted in Ningbo, China (*Zhu et al., 2024a*; *Zhu et al., 2024b*). Ningbo, located on the eastern coast of China, is one of the most economically developed cities, with a resident population of 9.697 million in 2023. Ningbo is situated in a subtropical monsoon climate zone characterized by distinct seasons. The average annual temperature is approximately 16.4 °C, the average annual sunshine duration is 1,850 h, and the average annual precipitation is 1,480 millimeters.

This study utilized clinical data from obstetric inpatients at four cooperative medical centers in Ningbo between January 1, 2013, and December 31, 2023. Under a collaborative framework approved by institutional review boards (ethical approval no. SX201916), standardized electronic medical records were consolidated through structured data abstraction protocols to establish a retrospective multicenter database for systematic analysis.

The inclusion criteria for all the subjects were as follows: aged 20–49 years, singleton pregnancies, natural conception, gestational week of at least 28 weeks, availability of diagnostic information for mothers, continuous residence during pregnancy, and Han Chinese ethnicity. Participants with pre-existing diagnoses of diabetes, cardiovascular disease, renal disorders, severe hepatic conditions, or cancer prior to pregnancy were excluded from the study. Pregnant individuals who experienced spontaneous or induced abortions during pregnancy or lacked documented last menstrual period (LMP) data before conception were also excluded. A total of 92,332 pregnant women were ultimately included in the analysis.

In our study, the minimum required sample size was calculated using the formula as following:

$$n = ((Z_{\alpha/2})^2 \cdot p \cdot (1-p))/E^2$$

where: $Z_{\alpha/2} = 1.96$: critical value for a two-tailed 95% confidence level ($\alpha = 0.05$); $p = 0.1976$: anticipated prevalence of GDM which derived from prior epidemiological study (*Qian et al., 2023*); $E = 0.02$: allowable margin of error ($\pm 2\%$).

This yielded a minimum requirement of 1,526 participants to ensure the estimated prevalence would fall within $\pm$ 2% of the true population value. Our study ultimately enrolled 92,332 participants, exceeding the minimum requirement by over 60-fold. This substantial sample size could minimize sampling error and enhance generalizability of findings to the target population.

Data collection encompassed demographic characteristics, medical records, and delivery details, systematically abstracted from hospital electronic medical records systems with de-identification of sensitive fields. Gestational weeks were determined by LMP date and ultrasound examination. Conception seasons (spring: March–May, summer: June–August, autumn: September–November and winter: December–February) were calculated based on LMP.

In accordance with the information security technology guide for health data security of China, sensitive personal information of all participants would undergo anonymization and structural modification through standardized encryption protocols to ensure irreversible privacy protection prior to any analytical processing occurs. This study was approved by the Ningbo University Medical Science Research Ethics Committee, as well as ethical consent from all collaborative medical centers (ethical application ref: SX201916). Furthermore, written informed consent for participation was not required due to the retrospective nature of this study, in accordance with Chinese national legislation and institutional requirements.

## Criteria for pregnancy complications

In this study, four common pregnancy complications were included in the analysis as outcomes: GDM, GH, PE, and hypothyroidism.

The diagnosis of GDM was performed using the standardized "one-step" 75-g oral glucose tolerance test (OGTT) according to the International Association of the Diabetes and Pregnancy Study Groups (IADPSG) criteria (*McIntyre et al., 2016*), with reference to the American Diabetes Association (ADA) diagnostic guidelines (*American Diabetes Association, 2012*). As previously described, the diagnostic procedure required: (1) administration of a 75-g glucose load after $\geq$ 8-hour overnight fasting at 24–28 gestational weeks; (2) glucose measurement at fasting, 1-hour and 2-hour intervals using standardized laboratory procedures. Diagnosis was confirmed when meeting any IADPSG/ADA threshold: fasting $\geq$ 5.1 mmol/L (92 mg/dL), 1-hour $\geq$ 10.0 mmol/L (180 mg/dL), or 2-hour $\geq$ 8.5 mmol/L (153 mg/dL).

GH and PE were diagnosed based on the 2021 guidelines issued by the International Society for the Study of Hypertension in Pregnancy (ISSHP) (*Magee et al., 2022*). GH is

new-onset hypertension at ≥20 weeks of gestation in the absence of proteinuria or other findings suggestive of preeclampsia. PE is gestational hypertension with ≥1 new onset of organ or uteroplacental dysfunction: proteinuria, other maternal end-organ dysfunction, or uteroplacental dysfunction.

The diagnostic criteria for hypothyroidism during pregnancy were established through adherence to the 2017 American Thyroid Association (ATA) Guidelines for Thyroid Disease Management in Pregnancy and Postpartum (*Alexander et al., 2017*). The diagnosis of hypothyroidism during pregnancy is based on the serum levels of thyroid-stimulating hormone (TSH) and free thyroxine (FT4). Clinical hypothyroidism (CH) is defined as a TSH level higher than the upper limit of the specific reference range (or 4.0 mU/L in the first trimester) and an FT4 level less than the lower limit of the specific reference range during pregnancy. Subclinical hypothyroidism (SCH) is defined as a TSH level higher than the upper limit of the specific reference range (or 4.0 mU/L in the first trimester) and an FT4 level in the normal range during pregnancy.

## Meteorological factor data

We collected the daily mean temperature ($T_{mean}$), relative humidity (RH), surface pressure, wind speed, precipitation, sunshine duration, maximum temperature ($T_{max}$), minimum temperature ($T_{min}$) and DTR in Ningbo from January 1, 2013, to December 31, 2023, to evaluate the relationships between meteorological factors and these four pregnancy complications. $T_{mean}$, RH, surface pressure, wind speed, precipitation, sunshine duration, $T_{max}$, and $T_{min}$ records were obtained from the National Aeronautics and Space Administration (NASA) Goddard Earth Sciences (GES) Data and Information Services Center (DISC) Global Land Data Assimilation System (GLDAS) (*Rodell et al., 2004*; *Beaudoing &.Rodell, 2020*). The DTR is calculated as the difference between $T_{max}$ and $T_{min}$.

We averaged these variables for the first day of each pregnancy week and the subsequent six days to assess the average individual exposure levels of meteorological factors per week of pregnancy for each participant. We calculated the average individual exposure levels of meteorological factors for the following three periods: the first trimester (1–13 weeks), the second trimester (14–27 weeks), and the first two trimesters combined (1–27 weeks).

## Definition of the MFS

In this study, we calculated the MFS according to these nine meteorological factors (*Wang et al., 2021*). Initially, individual exposure values for the nine meteorological factors across all participants were normalized using min–max standardization. The equation:

$X_{new} = (X - X_{min})/(X_{max} - X_{min})$.

We subsequently summed the exposure values of the nine meteorological factors and weighted them according to adjusted multivariate estimates of high-frequency risk (β coefficients) to derive the weighted MFS. The equation was as follows:

$MFS = (\beta_{T_{mean}} \times T_{mean} + \beta_{RH} \times RH + \beta_{surface\ pressure} \times surface\ pressure + \beta_{wind\ speed} \times wind\ speed + \beta_{precipitation} \times precipitation + \beta_{sunshine\ duration} \times sunshine\ duration +$

$\beta_{T_{max}} \times T_{max} + \beta_{T_{min}} \times T_{min} + \beta_{DTR} \times DTR) \times (9/\text{sum of the } \beta \text{ coefficients}).$

A higher MFS indicates a greater degree of exposure to environmental meteorological factors. In addition, the participants were divided into five groups according to the quintiles of the MFS.

## Statistical analysis

We used descriptive statistics to summarize the demographic characteristics of the study population and the exposure to meteorological factors during the study period. Continuous variables are presented as the means and standard deviations (means $\pm$ SDs), and categorical variables are presented as frequencies and percentages. The Pearson linear correlation test was employed to examine the relationships between the various meteorological factors. We adjusted all the models for the same confounding factors, including maternal age, parity (1, 2, $\geq$3), gravidity (1, $\geq$2), season of conception (spring, summer, autumn, or winter), and year of conception. The results for all the models are presented as odds ratios (ORs) with 95% confidence intervals (CIs).

We assessed the impact of each individual meteorological factor and MFS on pregnancy complications. Because GDM, GH and PE were mainly diagnosed in the second trimester in the study population, we used a logistic regression model to estimate the associations between each meteorological factor and MFS on the risk of GDM, GH and PE during the first trimester, the second trimester and the first two trimesters. As hypothyroidism is diagnosed mainly in the first trimester, we used a logistic regression model to estimate the associations between each meteorological factor and MFS with hypothyroidism during the first trimester.

We then assessed the effects of sensitive windows of exposure to extreme meteorological factors on the risk of pregnancy complications. Distributed lag nonlinear models (DLNMs) incorporating logistic regression were applied to estimate the exposure$-$lag$-$response effects of exposure to extreme meteorological factors on the risk of pregnancy complications (*Gasparrini, 2014*). The 95th, 97th and 99th percentiles of each meteorological variable were defined as extremely high values, whereas the 5th, 3rd and 1st percentiles were defined as extremely low values, with the median of each meteorological variable serving as the reference value (*Zhang et al., 2021*). Considering that GDM is diagnosed between weeks 24 and 28 of pregnancy and that GH and PE are diagnosed after week 20 of pregnancy (*McIntyre et al., 2016*; *Magee et al., 2022*), the lag time was set from 1–24 gestational weeks. Hypothyroidism is diagnosed early in pregnancy (*Alexander et al., 2017*), so the lag time range was set from 1–13 gestational weeks. Given the potential nonlinear effects of meteorological factors, we employed natural cubic spline functions to model the delayed response relationship between exposure at each gestational week and the outcome (*Wang et al., 2020*). The optimal degrees of freedom were determined based on the minimum Akaike information criterion (AIC) (*Gasparrini, 2014*).

Finally, we evaluated the effects of extreme meteorological factors during different stages of pregnancy on the risk of pregnancy complications. Meteorological factors were categorized into three groups. The 95th, 97th, and 99th percentiles of each meteorological variable were defined as extremely high values, whereas the 5th, 3rd, and 1st percentiles

were defined as extremely low values. The ranges from the 5th–95th percentiles, 3rd–97th percentiles, and 1st–99th percentiles were designated as reference values. Logistic regression models were used to assess whether extreme meteorological factors, compared with reference values, influenced the occurrence of pregnancy complications. For the interaction analysis, we used the relative risk owing to interaction (RERI), proportion attributable (AP) and synergy index (SI) to estimate the additive interactive effects of extreme meteorological factors on the risk of pregnancy complications in different trimesters (*Lou et al., 2018*; *Zhang et al., 2021*). Each statistically significant extreme meteorological factor was explored for its two-by-two additive interaction. If there is an additive interaction between the two factors, the RERI with 95% CI and AP with 95% CI should not include 0, whereas the SI with 95% CI should not include 1. RERI and AP > 0 and SI > 1 indicate positive interactions, whereas RERI and AP < 0 and SI > 1 indicate negative interactions.

All the statistical analyses were conducted using R software (version 4.2.3). A two-sided test with $P < 0.05$ was considered statistically significant. Each reported odds ratio in this study was the adjusted OR (aOR).

## RESULTS

### Basic characteristics of the participants and exposure to meteorological factors during pregnancy

A total of 92,332 participants were included in the study, with diagnoses as follows: gestational diabetes mellitus (GDM) in 17,814 participants (19.29%), gestational hypertension (GH) in 3,860 participants (4.18%), preeclampsia (PE) in 3,101 participants (3.36%), and hypothyroidism in 17,418 participants (18.86%) (Table 1). Overall, the mean (SD) maternal age of the participants was 30.02 (4.57) years, and 49,248 (53.34%) were primiparous. In general, advanced maternal age, increased gravidity and spring pregnancy increased the risk of GDM, GH and PE (all $P < 0.001$). Primiparous individuals had increased risks of GH, PE and hypothyroidism (all $P < 0.01$). Residents had increased risks of GDM, whereas immigrants had increased risks of GH and PE (all $P < 0.001$) (Tables S1–S4).

Table S5 presents the average exposure levels of meteorological factors during different trimesters in patients with GDM, GH, PE and hypothyroidism and in healthy pregnant women. During the first trimester, patients with GDM, GH, PE and hypothyroidism had higher exposure levels of $T_{mean}$, RH, precipitation, $T_{max}$ and $T_{min}$ and lower exposure levels of surface pressure (all $P < 0.001$). During the second trimester, patients with GDM, GH and PE exhibited significantly elevated exposure levels for $T_{mean}$, RH, $T_{max}$, and $T_{min}$ (all $P < 0.02$).

The correlations among the meteorological factors are shown in Fig. S1. Table S6 summarizes the trimester-specific distributions of meteorological factors across all participants. Weekly distributions of meteorological factors during the first 24 and 13 gestational weeks are detailed in Tables S7 and S8, respectively.

**Table 1 Maternal characteristics of all participants.**

| | All (n = 92,332) | GDM (n = 17,814) | GH (n = 3,860) | PE (n = 3,101) | Hypothyroidism (n = 17,418) |
|---|---|---|---|---|---|
| Maternal age (years, mean ± SD) | 30.02 ± 4.57 | 31.52 ± 4.72 | 30.80 ± 5.14 | 30.90 ± 5.21 | 30.05 ± 4.62 |
| Gravidity (n, %) | | | | | |
| 1 | 32,044 (34.71) | 5,312 (29.82) | 1,412 (36.58) | 1,062 (34.25) | 6,224 (35.73) |
| 2 | 25,137 (27.22) | 4,778 (26.82) | 944 (24.46) | 699 (22.54) | 4,570 (26.24) |
| ≥3 | 35,151 (38.07) | 7,724 (43.36) | 1,504 (38.96) | 1,340 (43.21) | 6,624 (38.03) |
| Parity (n, %) | | | | | |
| Primiparous | 49,248 (53.34) | 8,721 (48.96) | 2,174 (56.32) | 1,726 (55.66) | 9,845 (56.52) |
| Multiparous | 43,084 (46.66) | 9,093 (51.04) | 1,686 (43.68) | 1,375 (44.34) | 7,573 (43.48) |
| Residence (n, %) | | | | | |
| Residents | 45,319 (49.08) | 8,963 (50.31) | 1,937 (50.18) | 1,424 (45.92) | 7,933 (45.54) |
| Immigrants | 47,013 (50.92) | 8,851 (49.69) | 1,923 (49.82) | 1,677 (54.08) | 9,485 (54.46) |
| Fetal gender (n, %) | | | | | |
| Male | 49,025 (53.10) | 9,451 (53.06) | 2,038 (52.80) | 1,543 (49.76) | 9,463 (54.33) |
| Female | 43,293 (46.88) | 8,359 (46.92) | 1,821 (47.18) | 1,558 (50.24) | 7,951 (45.65) |
| Missing | 14 (0.02) | 4 (0.02) | 1 (0.02) | 0 | 4 (0.02) |
| Season of conception (n, %) | | | | | |
| Spring (March–May) | 22,632 (24.51) | 4,740 (26.61) | 1,121 (29.04) | 884 (28.51) | 4,291 (24.64) |
| Summer (June–August) | 21,289 (23.06) | 4,310 (24.20) | 971 (25.16) | 723 (23.31) | 4,354 (25.00) |
| Fall (September–November) | 23,414 (25.36) | 4,055 (22.76) | 779 (20.18) | 713 (22.99) | 4,385 (25.17) |
| Winter (December–February) | 24,997 (27.07) | 4,709 (26.43) | 989 (25.62) | 781 (25.19) | 4,388 (25.19) |

Notes.
GDM, gestational diabetes mellitus; GH, gestational hypertension; PE, preeclampsia; SD, standard deviation.

## Associations between meteorological factors and pregnancy complications

Table 2 summarizes trimester-specific associations between meteorological factors and pregnancy complications after adjusting for confounders. During the first trimester, higher temperatures ($T_{mean}$, $T_{max}$, $T_{min}$) consistently increased risks of GDM, GH, PE, and hypothyroidism. Increased precipitation was linked to higher GH and hypothyroidism risks, while prolonged sunshine duration raised the odds of GDM and GH. Conversely, rising surface pressure reduced risks of all complications. Higher RH increased GDM and PE risks but decreased hypothyroidism likelihood, whereas wind speed reduced PE risk but heightened hypothyroidism risk. In the second trimester, elevated temperatures continued to amplify GDM and GH risks. Higher RH remained positively associated with GDM and PE, and increased precipitation was linked to higher GDM risk, while increased surface pressure and DTR reduced GH and PE risks, respectively. All reported associations were statistically significant ($P < 0.05$), with detailed aORs and 95% CIs shown in Table 3.

Analysis between MFS and pregnancy complications revealed dose–response relationships. Participants in the highest MFS quintile during the first trimester had greater odds of GDM (aOR = 1.186, 95% CI [1.079–1.304]), GH (aOR = 1.596, 95% CI [1.323–1.925]), PE (aOR = 1.347, 95% CI [1.094–1.658]) and hypothyroidism (aOR = 1.257, 95% CI [1.141–1.385]) compared to the lowest quintile. In the second trimester,

**Table 2  Associations between meteorological factors with pregnancy complications in different trimesters among participants.**

| | | The first trimester | | The second trimester | | The first two trimesters | |
|---|---|---|---|---|---|---|---|
| | | OR (95% CI) | p-value | OR (95% CI) | p-value | OR (95% CI) | p-value |
| GDM | $T_{mean}$ (per 10 °C increase) | 1.110 (1.050, 1.174) | <0.001 | 1.074 (1.014, 1.138) | 0.014 | 1.203 (1.108, 1.306) | <0.001 |
| | RH (per 10% increase) | 1.058 (1.007, 1.112) | 0.025 | 1.054 (1.001, 1.110) | 0.045 | 1.145 (1.057, 1.240) | <0.001 |
| | Surface pressure (per 10 hPa increase) | 0.895 (0.848, 0.944) | <0.001 | 0.970 (0.919, 1.025) | 0.283 | 0.859 (0.793, 0.931) | <0.001 |
| | Wind speed (per 1 m/s increase) | 0.954 (0.878, 1.038) | 0.273 | 1.053 (0.955, 1.160) | 0.301 | 0.992 (0.855, 1.151) | 0.915 |
| | Precipitation (per 1 mm increase) | 1.001 (0.990, 1.013) | 0.801 | 1.014 (1.003, 1.026) | 0.014 | 1.019 (1.002, 1.037) | 0.032 |
| | Sunshine duration (per 1 h increase) | 1.093 (1.050, 1.138) | <0.001 | 0.993 (0.953, 1.035) | 0.729 | 1.096 (1.031, 1.166) | 0.003 |
| | $T_{max}$ (per 10 °C increase) | 1.110 (1.056, 1.167) | <0.001 | 1.061 (1.008, 1.117) | 0.023 | 1.188 (1.104, 1.280) | <0.001 |
| | $T_{min}$ (per 10 °C increase) | 1.107 (1.053, 1.165) | <0.001 | 1.070 (1.016, 1.126) | 0.011 | 1.195 (1.109, 1.287) | <0.001 |
| | DTR (per 10 °C increase) | 1.011 (0.987, 1.036) | 0.375 | 0.981 (0.955, 1.008) | 0.161 | 0.990 (0.946, 1.035) | 0.654 |
| GH | $T_{mean}$ (per 10 °C increase) | 1.182 (1.060, 1.319) | 0.003 | 1.316 (1.177, 1.472) | <0.001 | 1.590 (1.354, 1.867) | <0.001 |
| | RH (per 10% increase) | 0.998 (0.907, 1.097) | 0.961 | 1.086 (0.984, 1.198) | 0.099 | 1.104 (0.949, 1.285) | 0.200 |
| | Surface pressure (per 10 hPa increase) | 0.798 (0.719, 0.885) | <0.001 | 0.832 (0.748, 0.925) | <0.001 | 0.638 (0.546, 0.746) | <0.001 |
| | Wind speed (per 1 m/s increase) | 1.131 (0.963, 1.328) | 0.133 | 0.920 (0.767, 1.102) | 0.366 | 1.080 (0.813, 1.435) | 0.595 |
| | Precipitation (per 1 mm increase) | 1.028 (1.006, 1.050) | 0.012 | 1.007 (0.985, 1.029) | 0.548 | 1.044 (1.008, 1.081) | 0.015 |
| | Sunshine duration (per 1 h increase) | 1.234 (1.141, 1.335) | <0.001 | 1.081 (0.997, 1.172) | 0.058 | 1.383 (1.227, 1.559) | <0.001 |
| | $T_{max}$ (per 10 °C increase) | 1.181 (1.070, 1.303) | <0.001 | 1.264 (1.143, 1.397) | <0.001 | 1.522 (1.317, 1.759) | <0.001 |
| | $T_{min}$ (per 10 °C increase) | 1.160 (1.050, 1.280) | 0.003 | 1.265 (1.144, 1.399) | <0.001 | 1.498 (1.296, 1.731) | <0.001 |
| | DTR (per 10 °C increase) | 1.047 (0.999, 1.098) | 0.056 | 0.998 (0.948, 1.050) | 0.927 | 1.073 (0.984, 1.171) | 0.110 |
| PE | $T_{mean}$ (per 10 °C increase) | 1.135 (1.006, 1.280) | 0.040 | 1.060 (0.936, 1.200) | 0.359 | 1.223 (1.020, 1.466) | 0.030 |
| | RH (per 10% increase) | 1.111 (1.003, 1.231) | 0.044 | 1.116 (1.003, 1.242) | 0.043 | 1.288 (1.096, 1.513) | 0.002 |
| | Surface pressure (per 10 hPa increase) | 0.885 (0.787, 0.995) | 0.041 | 0.989 (0.879, 1.112) | 0.848 | 0.864 (0.724, 1.030) | 0.104 |
| | Wind speed (per 1 m/s increase) | 0.840 (0.709, 0.995) | 0.044 | 0.926 (0.767, 1.119) | 0.429 | 0.701 (0.520, 0.944) | 0.019 |
| | Precipitation (per 1 mm increase) | 1.018 (0.993, 1.043) | 0.158 | 1.010 (0.984, 1.036) | 0.450 | 1.036 (0.995, 1.078) | 0.088 |
| | Sunshine duration (per 1 h increase) | 1.072 (0.981, 1.170) | 0.123 | 0.940 (0.860, 1.027) | 0.170 | 1.003 (0.877, 1.147) | 0.962 |
| | $T_{max}$ (per 10 °C increase) | 1.131 (1.014, 1.260) | 0.027 | 1.038 (0.928, 1.160) | 0.514 | 1.189 (1.010, 1.399) | 0.037 |
| | $T_{min}$ (per 10 °C increase) | 1.133 (1.017, 1.263) | 0.024 | 1.071 (0.958, 1.197) | 0.230 | 1.235 (1.049, 1.453) | 0.011 |
| | DTR (per 10 °C increase) | 0.994 (0.944, 1.047) | 0.827 | 0.925 (0.874, 0.978) | 0.006 | 0.885 (0.806, 0.971) | 0.010 |
| Hypothyroidism | $T_{mean}$ (per 10 °C increase) | 1.145 (1.082, 1.212) | <0.001 | | | | |
| | RH (per 10% increase) | 0.911 (0.867, 0.957) | <0.001 | | | | |
| | Surface pressure (per 10 hPa increase) | 0.938 (0.889, 0.990) | 0.021 | | | | |
| | Wind speed (per 1 m/s increase) | 1.211 (1.113, 1.319) | <0.001 | | | | |
| | Precipitation (per 1 mm increase) | 1.013 (1.002, 1.024) | 0.024 | | | | |
| | Sunshine duration (per 1 h increase) | 0.991 (0.952, 1.031) | 0.646 | | | | |
| | $T_{max}$ (per 10 °C increase) | 1.092 (1.038, 1.149) | <0.001 | | | | |
| | $T_{min}$ (per 10 °C increase) | 1.101 (1.046, 1.159) | <0.001 | | | | |
| | DTR (per 10 °C increase) | 0.985 (0.961, 1.009) | 0.218 | | | | |

**Notes.**
OR, odds ratio; 95% CI, 95% confidence interval; $T_{mean}$, daily mean temperature; RH, relative humidity; $T_{max}$, daily maximum temperature; $T_{min}$, daily minimum temperature; DTR, diurnal temperature range; GDM, gestational diabetes mellitus; GH, gestational hypertension; PE, preeclampsia.
All models were adjusted for maternal age, gravidity, parity, season of conception and year of conception.

**Table 3  Associations between meteorological factor score (quintiles) with pregnancy complications in different trimesters among participants.**

| Pregnancy complications | | The first trimester | | The second trimester | | The first two trimesters | |
|---|---|---|---|---|---|---|---|
| | | OR (95% CI) | P-value for trend | OR (95% CI) | P-value for trend | OR (95% CI) | P-value for trend |
| GDM | Q1 | 1.000 | <0.001 | 1.000 | <0.001 | 1.000 | <0.001 |
| | Q2 | 1.040 (0.984, 1.099) | | 1.022 (0.967, 1.079) | | 1.054 (0.974, 1.141) | |
| | Q3 | 1.070 (1.005, 1.139) | | 1.070 (1.000, 1.146) | | 1.125 (1.021, 1.240) | |
| | Q4 | 1.137 (1.035, 1.250) | | 1.145 (1.049, 1.251) | | 1.184 (1.072, 1.307) | |
| | Q5 | 1.186 (1.079, 1.304) | | 1.124 (1.026, 1.231) | | 1.221 (1.078, 1.383) | |
| GH | Q1 | 1.000 | <0.001 | 1.000 | <0.001 | 1.000 | <0.001 |
| | Q2 | 1.066 (0.951, 1.193) | | 1.170 (1.051, 1.303) | | 1.005 (0.853, 1.184) | |
| | Q3 | 1.180 (1.040, 1.338) | | 1.198 (1.046, 1.372) | | 1.152 (0.944, 1.406) | |
| | Q4 | 1.400 (1.159, 1.690) | | 1.255 (1.046, 1.506) | | 1.355 (1.109, 1.655) | |
| | Q5 | 1.596 (1.323, 1.925) | | 1.382 (1.151, 1.659) | | 1.413 (1.101, 1.813) | |
| PE | Q1 | 1.000 | <0.001 | 1.000 | 0.021 | 1.000 | 0.024 |
| | Q2 | 1.010 (0.892, 1.144) | | 0.896 (0.787, 1.020) | | 1.129 (0.985, 1.293) | |
| | Q3 | 1.162 (1.010, 1.337) | | 0.876 (0.766, 1.003) | | 1.078 (0.915, 1.271) | |
| | Q4 | 1.170 (0.954, 1.434) | | 0.826 (0.710, 0.962) | | 1.220 (1.025, 1.452) | |
| | Q5 | 1.347 (1.094, 1.658) | | 0.800 (0.673, 0.952) | | 1.248 (1.010, 1.542) | |
| hypothyroidism | Q1 | 1.000 | <0.001 | | | | |
| | Q2 | 1.097 (1.038, 1.159) | | | | | |
| | Q3 | 1.116 (1.047, 1.189) | | | | | |
| | Q4 | 1.252 (1.139, 1.377) | | | | | |
| | Q5 | 1.257 (1.141, 1.385) | | | | | |

**Notes.**

OR, odds ratio; 95% CI, 95% confidence interval; MFS, meteorological factor score; GDM, gestational diabetes mellitus; GH, gestational hypertension; PE, preeclampsia; SD, standard deviation.
All models were adjusted for maternal age, gravidity, parity, season of conception and year of conception.

high MFS quintile exposure increased GDM and GH risks but decreased PE likelihood. Cumulative exposure across the first two trimesters further amplified GDM, GH, and PE risks (Table 3). All trend tests reached statistical significance ($P < 0.05$), with consistent patterns observed per 10-point MFS increment (Table S9). These findings indicated that cumulative exposure to adverse meteorological conditions during the first and second trimesters elevated complication risks in a dose-dependent manner.

## Effects of extreme meteorological factors on risks of pregnancy complications

Tables S12–S15 demonstrate trimester-specific associations between extreme meteorological factors and pregnancy complications risks. Figures 1 and 2 identifies susceptibility windows for extremely high meteorological exposures on risks of GDM, GH and PE during the first 24 gestational weeks and hypothyroidism during the first 13 gestational weeks. Compared with the median meteorological exposure, extremely high precipitation and sunshine duration elevated GDM risks during 13th–20th and 1st–8th gestational weeks, respectively. For GH, elevated risks clustered between during 15th–23rd weeks with exposure to extremely high $T_{mean}$, RH, wind speed, precipitation, sunshine, and $T_{max}$. PE risks increased during early (1st–10th weeks) and late (18th–24th weeks) gestation under

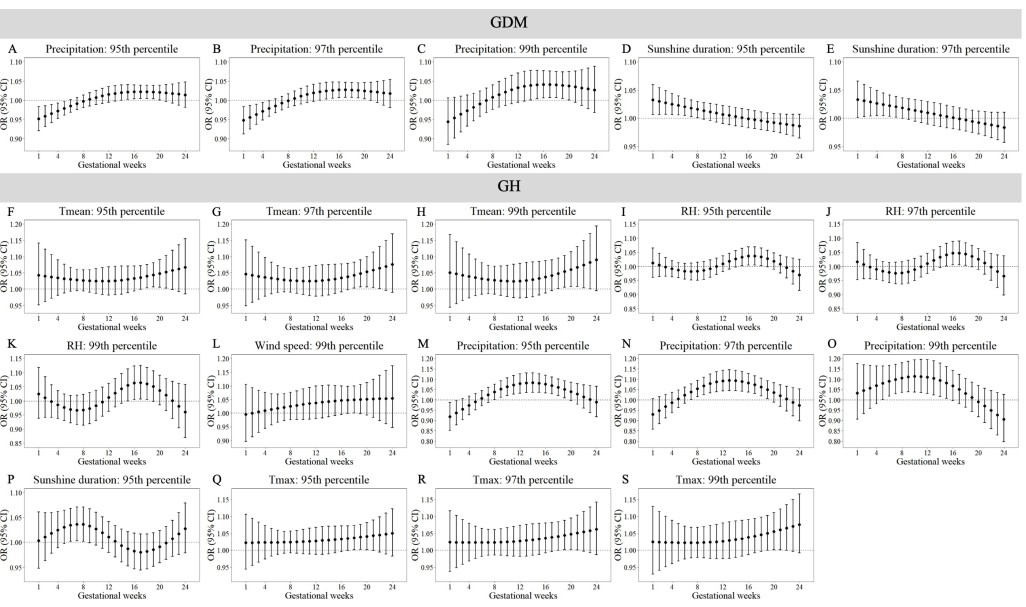

**Figure 1** **Susceptibility windows for extremely high meteorological exposures on risks of GDM and GH.** Maternal weekly-specific extremely high meteorological exposures on risks of GDM and GH were depicted by distributed lag non-linear models. Extremely high meteorological factors were defined by different percentiles (95th, 97th and 99th) of meteorological factors. All models were adjusted for maternal age, gravidity, parity, season of conception and year of conception. GDM, gestational diabetes; GH, gestational hypertension.

extremely high RH, surface pressure, precipitation, sunshine, and $T_{max}$. Hypothyroidism showed heightened sensitivity to multiple extremely high factors ($T_{mean}$, RH, surface pressure, wind speed, precipitation, $T_{max}$, $T_{min}$, DTR) predominantly in the first trimester (1st–13th weeks).

Figures 3 and 4 reveals susceptibility windows for extremely low meteorological exposures. Compared with the median meteorological factors, extremely low DTR increased GDM risk during the 1st gestational week. GH risks rose during 9th–19th weeks with extremely low surface pressure, wind speed, and $T_{min}$. PE risks were elevated during 10th–21st weeks under low precipitation, $T_{max}$, and DTR. Hypothyroidism exhibited broad sensitivity to extremely low meteorological values ($T_{mean}$, RH, surface pressure, wind speed, precipitation, $T_{max}$, $T_{min}$, DTR), primarily within the first trimester (1st–13th weeks).

Non-significant associations are detailed in Tables S10–S11. In summary, extreme meteorological exposures influenced complications within distinct windows: risks of GDM, GH, and PE concentrated in mid-pregnancy (3rd–5th months), while hypothyroidism showed first-trimester vulnerability.

## Effects of interactions between extreme meteorological factors on risks of pregnancy complications

Analysis of potential additive interactions between extreme meteorological factors revealed no statistically significant effects on risks of GDM, GH and PE (Tables S16–S18). In

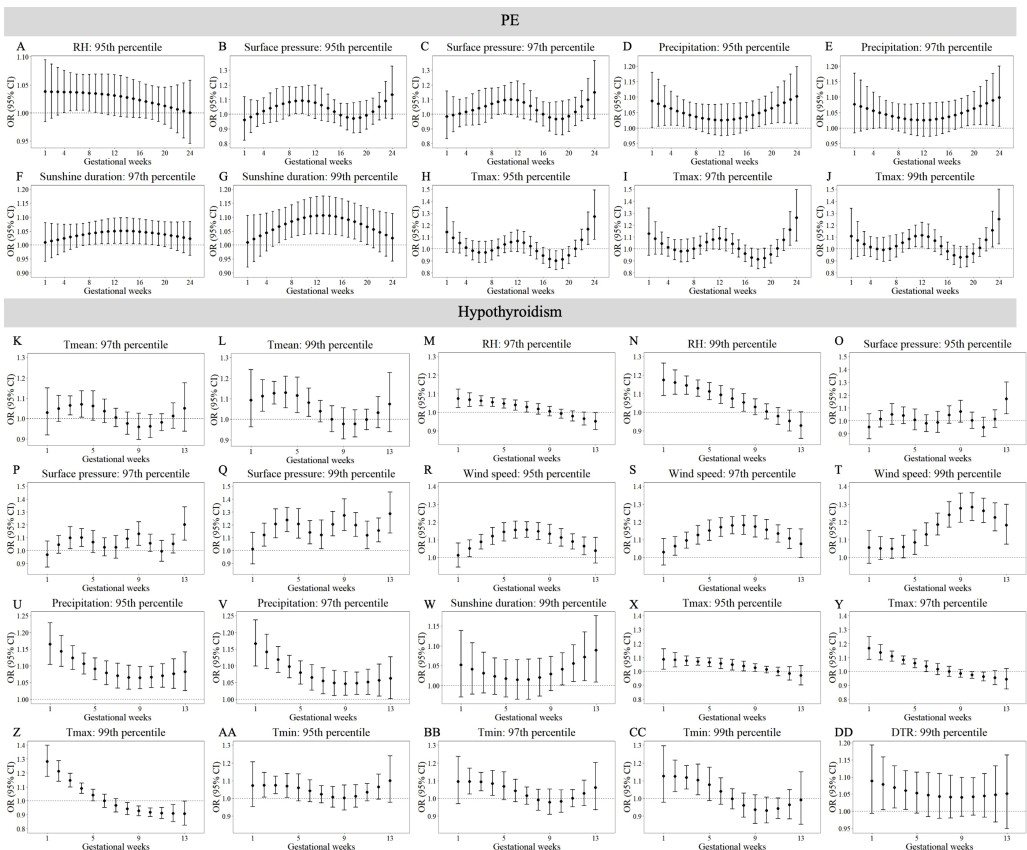

**Figure 2  Susceptibility windows for extremely high meteorological exposures on risks of PE and hypothyroidism.** Maternal weekly-specific extremely high meteorological exposures on risks of PE and hypothyroidism were depicted by distributed lag non-linear models. Extremely high meteorological factors were defined by different percentiles (95th, 97th and 99th) of meteorological factors. All models were adjusted for maternal age, gravidity, parity, season of conception and year of conception. PE, preeclampsia.

contrast, Table S19 demonstrates significant interactions between these factors on the risk of hypothyroidism during the first trimester.

Significant interactions between extreme meteorological factors on hypothyroidism risk were identified under varying percentile thresholds (Table 4). When extremes were defined as the 5th/95th percentiles, positive interactions occurred between extremely low $T_{min}$ (3.97 °C) and low precipitation (2.36 mm), high wind speed (4.03 m/s), or high DTR (8.93 °C), as well as between high wind speed and high DTR (all RERI > 0.60). For the 3rd/97th percentile thresholds, positive interactions emerged between extremely low $T_{mean}$ (7.74 °C) and high wind speed (4.17 m/s; RERI = 3.02) or high DTR (9.07 °C; RERI = 2.53), while a negative interaction was observed between low precipitation (2.13 mm) and high wind speed (RERI = −0.73). Under the 1st/99th percentile definition, a positive interaction linked low $T_{mean}$ (7.44 °C) and high surface pressure (1,023.13 hPa; RERI = 2.02), whereas negative interactions involved low RH (65.32%) with low $T_{min}$ (2.76 °C; RERI = −1.66) or high surface pressure (RERI = −1.09). All interaction metrics (RERI,

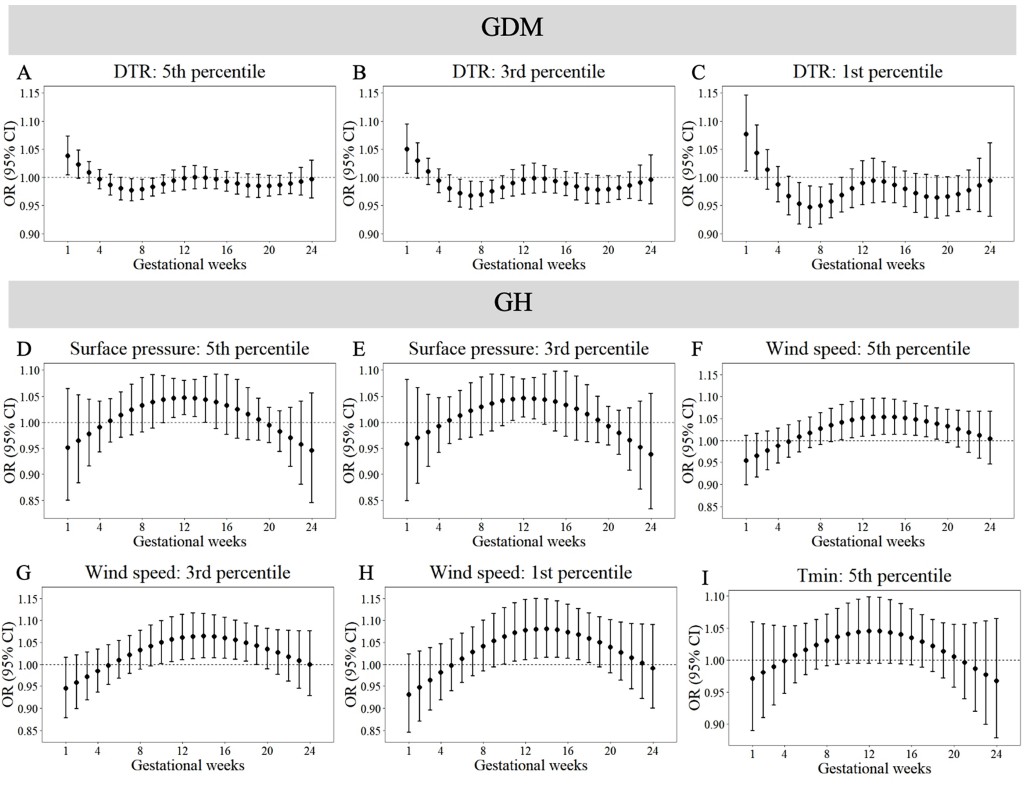

**Figure 3 Susceptibility windows for extremely low meteorological exposures on risks of GDM and GH.** Maternal weekly-specific extremely low meteorological exposures on risks of GDM and GH were depicted by distributed lag non-linear models. Extremely low meteorological factors were defined by different percentiles (5th, 3rd and 1st) of meteorological factors. All models were adjusted for maternal age, gravidity, parity, season of conception and year of conception. GDM, gestational diabetes; GH, gestational hypertension.

AP, SI) with 95% CIs are detailed in Table 4. No additional significant interactions were detected (Table S19).

## DISCUSSION

Our study systematically explored the impacts of external meteorological conditions on the reproductive health of pregnant women. In this study, we not only evaluated the independent risk effects and additive interactions of various meteorological factors during the first and second trimesters in terms of pregnancy complications but also constructed an MFS based on multidimensional meteorological factors to comprehensively assess the associations between the overall level of exposure to meteorological factors during pregnancy and pregnancy complications. We also assessed the sensitive time windows for the effects of extreme meteorological factors on the risks of different pregnancy complications.

The effects of meteorological factors on pregnant women have been previously reported in the literature. *Booth et al. (2017)* reported that each 10 °C increase in the mean 30-day

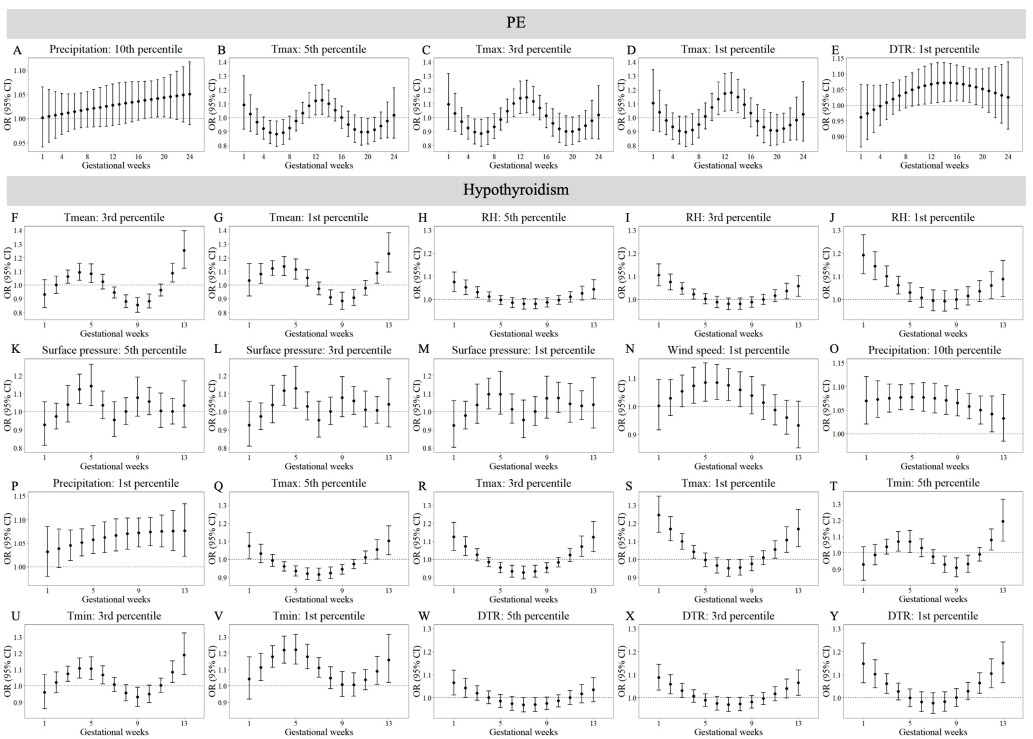

**Figure 4  Susceptibility windows for extremely low meteorological exposures on risks of PE and hypothyroidism.** Maternal weekly-specific extremely low meteorological exposures on risks of PE and hypothyroidism were depicted by distributed lag non-linear models. Extremely low meteorological factors were defined by different percentiles (5th, 3rd and 1st) of meteorological factors. All models were adjusted for maternal age, gravidity, parity, season of conception and year of conception. PE, preeclampsia.

temperature before the 27th week of pregnancy increased the GDM risk by 6%. *Su et al. (2020)* reported that GDM risk increased by 3% for each 1 °C increase in the mean 35-day period prior to GDM diagnosis from 14 °C to 27 °C, whereas GDM risk increased by 54% at temperatures above 28 °C. A higher prevalence of GDM was associated with a smaller difference in temperature within a day. Some studies have reported that extreme meteorological factors are also linked to GDM risk. *Teyton et al. (2023)* reported that GDM risk increased with extremely low temperatures from the 20th–24th gestational weeks and with extremely high temperatures from the 11th–14th gestational weeks. *Zhang et al. (2021)* reported that the time window of susceptibility to the effects of extreme temperatures and DTR on GDM occurred in the second trimester. Our findings are partially consistent with these studies, which revealed that not only mean temperature but also other meteorological factors, including RH, surface pressure, precipitation, sunshine duration, $T_{max}$ and $T_{min}$, were associated with the risk of GDM during the first and second trimesters in the present study. However, there was no association between extreme temperature and GDM risk in this study. In our study, GDM risk increased with extremely high precipitation from the 13th–20th gestational weeks, sunshine duration from the 1st–8th gestational weeks, and extremely low DTR during the 1st gestational week.

**Table 4   Interactive analysis between extreme meteorological factors on the risk of hypothyroidism in the first trimester.**

| Meteorological factors | | | Interaction effect | RERI (95% CI) | AP (95% CI) | SI (95% CI) |
|---|---|---|---|---|---|---|
| Extremely low and high meteorological factors are defined by 5th and 95th percentile of meteorological factors, respectively | | | | | | |
| Extremely low precipitation (2.36 mm) | and | Extremely low $T_{min}$ (3.97 °C) | Positive interaction | 0.60 (0.23, 1.03) | 0.33 (0.13, 0.45) | 3.89 (1.68, 9.00) |
| Extremely low $T_{min}$ (3.97 °C) | and | Extremely high wind speed (4.03 m/s) | Positive interaction | 1.06 (0.60, 1.61) | 0.44 (0.27, 0.54) | 3.90 (2.00, 7.61) |
| Extremely low $T_{min}$ (3.97 °C) | and | Extremely high DTR (8.93 °C) | Positive interaction | 0.74 (0.33, 1.22) | 0.39 (0.19, 0.50) | 5.41 (1.94, 15.07) |
| Extremely high wind speed (4.03 m/s) | and | Extremely high DTR (8.93 °C) | Positive interaction | 0.66 (0.29, 1.05) | 0.36 (0.17, 0.49) | 4.67 (1.14, 19.09) |
| Extremely low and high meteorological factors are defined by 3rd and 97th percentile of meteorological factors, respectively | | | | | | |
| Extremely low $T_{mean}$ (7.74 °C) | and | Extremely high wind speed (4.17 m/s) | Positive interaction | 3.02 (1.45, 5.36) | 0.65 (0.42, 0.74) | 5.89 (2.88, 12.03) |
| Extremely low $T_{mean}$ (7.74 °C) | and | Extremely high DTR (9.07 °C) | Positive interaction | 2.53 (0.69, 6.17) | 0.67 (0.23, 0.72) | 12.45 (3.61, 42.91) |
| Extremely low precipitation (2.13 mm) | and | Extremely high wind speed (4.17 m/s) | Negative interaction | −0.73 (−1.45, −0.10) | −0.48 (−1.10, −0.08) | 0.42 (0.19, 0.92) |
| Extremely low and high meteorological factors are defined by 1st and 99th percentile of meteorological factors, respectively | | | | | | |
| Extremely low $T_{mean}$ (7.44 °C) | and | Extremely high surface pressure (1,023.13 hPa) | Positive interaction | 2.02 (0.27, 5.07) | 0.54 (0.03, 0.68) | 3.93 (1.36, 11.37) |
| Extremely low RH (65.32%) | and | Extremely low $T_{min}$ (2.76 °C) | Negative interaction | −1.66 (−2.69, −0.82) | −1.10 (−2.05, −0.50) | 0.23 (0.10, 0.53) |
| Extremely low RH (65.32%) | and | Extremely high surface pressure (1,023.13 hPa) | Negative interaction | −1.09 (−2.2, −0.18) | −0.63 (−1.48, −0.12) | 0.40 (0.18, 0.89) |

**Notes.**

RERI, relative risk owing to interaction; AP, proportion attributable; SI, synergy index; 95% CI, 95% confidence interval; $T_{mean}$, daily mean temperature; RH, relative humidity; $T_{min}$, daily minimum temperature; DTR, diurnal temperature range.

Extreme meteorological factors were defined by different percentiles (5th, 3rd, 1st and 95th, 97th, 99th) of meteorological factors. All models were adjusted for maternal age, gravidity, parity, season of conception and year of conception.

There are several biological hypotheses to explain the influence of meteorological factors on the risk of GDM. Brown adipose tissue (BAT) plays an important role in glucose metabolism. Cold exposure can increase BAT activation, which improves whole-body glucose homeostasis and insulin sensitivity (*Van der Lans et al., 2013*; *Chondronikola et al., 2014*). High-temperature exposure reduces insulin sensitivity. High temperatures may also lead to beta-cell dysfunction in pregnant women, which reduces insulin sensitivity (*Retnakaran et al., 2018*). *Moses et al. (1997)* reported the acute effect of ambient temperature on apparent glucose tolerance and concluded that this effect was due to changes in core body temperature leading to the redistribution of blood flow between the cutaneous and visceral beds. Adequate vitamin D levels may reduce the risk of GDM (*Rizzo et al., 2019*). Skin exposure to solar ultraviolet B radiation is a major source of vitamin D (*Saraff & Shaw, 2016*). High ambient temperatures, high precipitation and low sunshine duration will make people go out less, reducing the opportunity to receive sunlight, leading to insufficient synthesis of vitamin D in the body. Furthermore, relative dehydration of

the body caused by high temperature leads to hemoconcentration, causing an elevation in blood glucose levels (*Preston et al., 2020*). The DTR is an indicator that can reflect weather stability. A review revealed that DTR is significantly associated with human mortality and morbidity, especially cardiovascular and respiratory diseases (*Cheng et al., 2014*). However, extremely low DTR was associated with an increased risk of GDM in our study. We believe that greater temperature fluctuations may increase insulin sensitivity; therefore, it is necessary to study the potential effects of temperature fluctuations on metabolic system diseases.

Previous studies have also shown associations between meteorological factors and the risks of GH and PE. *Xiong et al. (2020)* reported that an increased $T_{mean}$ during the first half of pregnancy increased the risk of PE or eclampsia and GH. Under extreme temperatures, extremely cold exposure during the first half of pregnancy decreased the risk of PE, eclampsia or GH, whereas extremely heat exposure increased the risk. *Zeng et al. (2024)* reported that extremely high temperatures (aOR = 1.24, 95% CI [1.12–1.38]) and moderately high temperatures (aOR = 1.22, 95% CI [1.10–1.35]) during the first trimester were associated with increased risks of PE. *Zhao, Long & Lu (2022)* reported that low-temperature exposure was a significant risk factor for preeclampsia risk. Overall, the results of the effects of temperature exposure on GH and PE risk were inconsistent. In our study, every 10 °C increase in $T_{mean}$, $T_{max}$ or $T_{min}$ during the first two trimesters was associated with an increased risk of GH. A 10 °C increase in $T_{mean}$, $T_{max}$ or $T_{min}$ during the first trimester was associated with an increased risk of PE. Under extreme temperatures, both extremely high and extremely low temperatures were associated with the risks of GH and PE. Extremely high temperatures increased the risks of GH and PE during the second trimester, whereas extremely low temperatures increased the risks of GH and PE mainly during the first trimester.

The mechanisms underlying the effects of temperature exposure on HDP risk are unclear. Some studies have suggested that the pathogenesis of preeclampsia or eclampsia and gestational hypertension are similar (*Xiong et al., 2020*). As pregnancy progresses, the increasing weight of the pregnant woman and the need for fetal growth lead to a decrease in maternal heat loss and an increase in internal heat production. Thus, heat exposure could disrupt thermoregulation and increase the risks of GH and PE (*Mao et al., 2023*). When exposed to cold temperatures, both the sympathetic nervous system and the renin–angiotensin system are activated, which could cause vasoconstriction and an increase in blood pressure (*Park et al., 2020*). In addition, cold exposure induces endothelial dysfunction. Thus, cold exposure is also a risk factor for HDP. It seems that the impact of extreme temperatures on HDP risk is bidirectional. Vitamin D benefits maternal health. Adequate vitamin D is important for reducing the risk of GDM, GH, PE and other complications (*Zhang et al., 2022*). In our study, extremely high RH, precipitation and sunshine increased the risks of both GH and PE. We believe that extremely high humidity, precipitation and sunshine duration lead to reduced outdoor activity, resulting in insufficient vitamin D production by the body. The observed discrepancies may stem from variations in exposure definitions across studies, inconsistent adjustment for confounding factors, heterogeneity in outcome definitions, differential population

susceptibility, and geographic disparities in meteorological patterns. Divergent approaches to classifying extreme temperatures (*e.g.*, percentile thresholds *vs.* absolute cutoffs) and distinct gestational window selections could further contribute to these inconsistencies. Our trimester-stratified, multi-factorial approach enhances granularity, revealing temporal and exposure-level subtle differences that broader analyses may overlook.

The thyroid gland is an important endocrine organ that regulates metabolism by secreting thyroid hormones (THs). TH production and release are regulated by the hypothalamic−pituitary−thyroid axis (*Chaker et al., 2022*). Thus, the thyroid gland is very sensitive to changes in ambient temperature. Few studies have investigated meteorological factors and the risk of hypothyroidism. In our study, the impact of extreme temperatures in the first trimester on the risk of hypothyroidism was bidirectional. Both extremely high and extremely low temperatures increased the risk of hypothyroidism. We believe that extreme temperatures affect the regulation of the hypothalamic−pituitary−thyroid axis. Almost all extreme meteorological factors are associated with an increased risk of hypothyroidism. Hypothyroidism may be more susceptible to meteorological factors in early pregnancy than other pregnancy complications. Vitamin D deficiency may be associated with an increased risk of thyroid autoimmunity (*Nettore et al., 2017*). Extreme weather can reduce people's ability to perform outdoor activities, which leads to insufficient synthesis of the sunshine vitamin.

Our study revealed that pregnant women were more vulnerable to meteorological factors in the first trimester than in the second trimester, which increased the risk of pregnancy complications. This is consistent with the conclusions reached by others (*Zeng et al., 2024*). Early pregnancy is a critical period for embryo implantation, vascularization and placentation (*Xiong et al., 2020*). Dramatic changes in hormone levels in pregnant women during early pregnancy and changes in the external environment, including changes in meteorological factors, may interfere with these processes and lead to complications during pregnancy (*Dreiling, Carman & Brown, 1991*; *Wang et al., 2024*).

With respect to extreme temperatures, both extremely high and extremely low meteorological factors increased the risk of pregnancy complications, but extremely high meteorological factors had greater impacts. GDM, GH, PE and hypothyroidism are all endocrine diseases that occur during pregnancy (*Amin, Robinson & Teoh, 2011*). Environmental heat exposure affects many hormones associated with adaptation to heat, including cortisol, thyroid hormones, arginine vasopressin (AVP), prolactin, growth hormone (GH), insulin, and adipokines. High temperature affects thermoregulation and contributes to the development of endocrine diseases by inducing a stress hormone response, inhibiting the activity of the thyroid axis, regulating the body's water balance, affecting sweating and heat loss, affecting insulin sensitivity, and affecting adipose tissue (*Hannan et al., 2024*). Thus, appropriate meteorological conditions may reduce the risk of pregnancy complications for pregnant women.

To our knowledge, this is the first study to use the MFS metric to assess its association with the risk of pregnancy complications. In our study, MFS was significantly associated with the risk of all four pregnancy complications. Our constructed MFS metrics were better predictors of the risk of pregnancy complications.

 

This study is the first to explore the effects of interactions between meteorological factors and the risk of pregnancy complications. We detected interactions between extreme meteorological factors and the risk of hypothyroidism, and the interactions were mainly between extreme temperature-related variables, including extremely low $T_{mean}$, extremely low $T_{min}$ and extremely high DTR, and other meteorological factors.

The impact of meteorology on the reproductive health of pregnant women is complex; therefore, we need to take a multifaceted and holistic view of its role. More epidemiologic and physiologic studies are needed to investigate the associations between temperature and other meteorological factors and pregnancy complications and their mechanisms of effects.

There are several limitations in our study. We were not able to obtain the specific diagnosis date of each pregnancy complication. We had no additional information on the pregnant women, such as pre-pregnancy BMI, prior history of gestational complications, household income, education, maternal alcohol consumption and maternal tobacco smoking, which left us with few covariates to adjust in the models. In addition, we had no specific home address information for pregnant women, which prevented us from accurately assessing the individual exposures of pregnant women. However, we believe that meteorological factors fluctuated little in our study area, which did not affect our results. In our study, the MFS, which is based on multidimensional meteorological factors, was used for the first time to assess the correlation between environmental meteorological factors and pregnancy complications comprehensively. Based on weekly exposure levels during pregnancy, we assessed the sensitive time window for the impact of extreme meteorological factors on complications during pregnancy. Our research comprehensively reflected the impact of climate change on pregnant women's health.

## CONCLUSIONS

Our study suggested that meteorological factors are critical risk factors for pregnancy complications. MFS was a better indicator for evaluating the associations between meteorological factors and pregnancy complications. Both extremely high and extremely low meteorological factors increased the risk of pregnancy complications. The sensitive time windows for extremely high and low meteorological factors on risks of GDM, GH and PE were mainly from the 3rd to 5th months of pregnancy, whereas the sensitive time window for extremely high and low meteorological factors on the risk of hypothyroidism was mainly throughout the first trimester. Interactions between meteorological factors collectively influenced the risk of hypothyroidism. Targeted protective strategies require multi-sector collaboration. Healthcare providers, employers, and policymakers should implement concrete measures: temperature-regulated workplaces for pregnant workers, weather-responsive prenatal care alerts, and heatwave-specific hydration guidance to mitigate environmental risks during vulnerable gestational periods.

## ACKNOWLEDGEMENTS

We thank the four collaborating medical centers for granting permission for data collection in this study and thank Ms. Yingying Li, Mr. Penghao Wang, and Mr. Liqi Xu for collecting the pregnant women's data of this research.

### Funding

This work was funded by Beijing Zhongwei Joint Funds of the Zhejiang Provincial Natural Science Foundation of China (No. LBY24H040001 and No. LBY24H040002), Zhejiang Medicine and Health Science and Technology Project (No. 2024XY149), Ningbo Youth Science and Technology Innovation Leaders Project (No. 2023QL057) and Ningbo Key Research and Development Project (No. 2024Z223 and No. 2024Z225). The funders had no role in study design, data collection and analysis, decision to publish, or preparation of the manuscript.

### Grant Disclosures

The following grant information was disclosed by the authors:
Beijing Zhongwei Joint Funds of the Zhejiang Provincial Natural Science Foundation of China: No. LBY24H040001 and No. LBY24H040002.
Zhejiang Medicine and Health Science and Technology Project: No. 2024XY149.
Ningbo Youth Science and Technology Innovation Leaders Project: No. 2023QL057.
Ningbo Key Research and Development Project: No. 2024Z223 and No. 2024Z225.

### Competing Interests

The authors declare there are no competing interests.

### Author Contributions

- Tao Chen conceived and designed the experiments, performed the experiments, analyzed the data, prepared figures and/or tables, authored or reviewed drafts of the article, and approved the final draft.
- Jie Cai conceived and designed the experiments, performed the experiments, analyzed the data, prepared figures and/or tables, authored or reviewed drafts of the article, and approved the final draft.
- Da He conceived and designed the experiments, prepared figures and/or tables, authored or reviewed drafts of the article, and approved the final draft.
- Hui Zhu analyzed the data, authored or reviewed drafts of the article, and approved the final draft.
- Xiang Chen performed the experiments, prepared figures and/or tables, and approved the final draft.
- Yuqiong Li performed the experiments, analyzed the data, prepared figures and/or tables, and approved the final draft.

- Wen Ye performed the experiments, prepared figures and/or tables, and approved the final draft.
- Bingqi Li performed the experiments, prepared figures and/or tables, and approved the final draft.
- Qinyan Xu conceived and designed the experiments, analyzed the data, prepared figures and/or tables, and approved the final draft.
- Lindan Ji conceived and designed the experiments, authored or reviewed drafts of the article, and approved the final draft.
- Jin Xu conceived and designed the experiments, authored or reviewed drafts of the article, and approved the final draft.

## Human Ethics

The following information was supplied relating to ethical approvals (i.e., approving body and any reference numbers):

This study was approved by the Ningbo University Medical Science Research Ethics Committee (Ethical Application Ref: SX201916).

## Data Availability

The raw measurements are available in the Supplemental Files.

## Supplemental Information

Supplemental information for this article can be found online at http://dx.doi.org/10.7717/peerj.19621#supplemental-information.

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
