# Peer review of "Associations between meteorological factors and pregnancy complications during different pregnancy trimesters: a multicenter retrospective study in eastern China"

_PeerJ, doi:10.7717/peerj.19621_

## Round 0.1 · original submission · Minor Revisions

Both reviewers have recommended minor revisions and provided thoughtful comments to support those revisions. These comments address both the clarity of the writing and specific technical issues. Please review the comments carefully and provide a detailed description of how each comment has been addressed in the revised manuscript.

**Language Note:** The review process has identified that the English language must be improved. PeerJ can provide language editing services - please contact us at [email protected] for pricing (be sure to provide your manuscript number and title). Alternatively, you should make your own arrangements to improve the language quality and provide details in your response letter. – PeerJ Staff

·

Basic reporting

The grammar and clarity of the entire manuscript should be revised and presented in a manner that makes it appealing to readers.
Abstract section:
The Results section of the abstract should be written clearly and concisely to engage readers (line numbers 35 to 56). For example, it is better to state: 'A total of 92,332 participants were included in the study, with diagnoses as follows: gestational diabetes mellitus (GDM) in 17,814 participants (19.29%), gestational hypertension (GH) in 3,860 participants (4.18%), preeclampsia (PE) in 3,101 participants (3.36%), and hypothyroidism in 17,418 participants (18.86%).
Introduction section:
The third paragraph (lines 102 to 106) does not include any references to the literature. Please provide the appropriate citations.
Result section:
The entire narration of result interpretation includes the odds ratio and confidence interval for each significant variable. Once you present the measure of association, such as the adjusted odds ratio (AOR) and confidence intervals for the variables in the tables, it is advisable to avoid repeating this information during the interpretation. Therefore, simply providing a narrative description of the direction of association is sufficient in the results interpretation.
You repeatedly used the phrase 'were associated with increased risks of...', which may come across as monotonous for readers. To enhance engagement, please consider using synonyms such as 'more likely', 'the odds of', or 'the likelihood of' instead.
Discussion section:
You should explain the possible reasons for any discrepancies or inconsistencies with previous studies (line number 439).

Experimental design

Materials & Methods section:
Total sample size has been mentioned, but you did not show how you determine it. Thus, please explain how you determine it. In addition, reference of diagnosis criteria for pregnancy complications is not cited (line no 154 to 160) . So, please cite the source.
Ethical consideration related question:
• You did not explain how you obtain ethical consent from participants and medical centers. Please include details on how to secure oral or written ethical consent, as well as the measures you take to maintain the confidentiality of participants' information.

Validity of the findings

In the conclusion section, you recommended that targeted protective measures should be implemented to reduce the risk of pregnancy complications. However, it is unclear who is responsible for carrying out these measures. Please clarify who holds this responsibility.

Additional comments

Acknowledgment issue:
In addition to thanking the data collectors, it is important to express gratitude to the medical center or organization that granted you permission to collect the data.

Reviewer 2 ·

Basic reporting

1. The sentence structure within the introduction is very basic and reads as a series of short statements, rather than a paragraph that flows from one concept to the next. I would encourage the authors to improve the sentence structure so that it flows nicely.
2. In the second paragraph of the introduction, the sentences read as choppy non-flowing statements, and I would encourage the authors to use more complex sentence structure to improve the flow of the paragraph for the reader.
3. The last paragraph of the results describing the effects of interactions is very difficult to read. I wonder if the authors could distill this down to a more readable format, maybe with only the most important numbers in the text and the rest in a table for the reader to refer to. It was easy to get lost in all of the numbers and not really understand the point of the paragraph.
4. Line 528: the statement “ interactions between …” is grammatically not correct. I think the authors mean to say “Interactions between extreme meteorological factors…”

Experimental design

1. Could the authors clarify about the exclusion criteria – were these diagnoses pre-pregnancy? (line 126)
2. The authors should re-define the abbreviations for the 4 complications in the first sentence on line 139 so that the reader doesn’t have to scroll up to the previous section
3. How did the authors collect the data for the participants? Was there a database used? Did they contact each hospital? I think it would be important to add a line about this
4. Did the authors have any data about previous pregnancy complications for those women who had previous pregnancies? If so, how did they adjust for this, as women who had previous complications are probably at risk for more?
5. Make sure there are references for all the claims about timing of diagnoses during pregnancy. The authors state that GDM is diagnosed between 24-28 weeks and other claims about timing of diagnosis for hypothyroidism, GE and PE so the authors need to have references for all of these claims.

Validity of the findings

1. Line 255-258: I would ask the authors to please put in p values to help the reader determine the significance of the statements you are making about higher vs lower levels. This will make it easier to follow as the reader is reading the paragraph without jumping separately to the table to see the significance.
2. The last paragraph of the results describing the effects of interactions is very difficult to read. I wonder if the authors could distill this down to a more readable format, maybe with only the most important numbers in the text and the rest in a table for the reader to refer to. It was easy to get lost in all of the numbers and not really understand the point of the paragraph.
3. Line 485 – need a reference for this statement.

Additional comments

I applaud the authors for a very comprehensive and interesting analysis with advanced statistical methods to investigate the association of meteorological and climate factors with pregnancy complications. This is an interesting paper that has compelling data to suggest that there may be an association between extreme environmental factors and some pregnancy complications. I think there are some limitations to the study, especially considering the authors did not have much data about overall maternal health and pre-pregnancy factors. While interesting and compelling, further study will be needed to determine the direct impact of meteorological factors on pregnancy, especially with case-control groups or exposure groups. Without this data comparing groups with and without the complications, as well as with and without the exposure factors, it is not possible to know for sure if these were risk factors, or merely associated. I have structured my comments in order of appearance in the manuscript.

1. In the introduction, I’m not sure that you can claim that meteorological factors are the “biggest” health threat, and I would recommend instead say something like “global meteorological factors have changed dramatically and have been correlated with a threat to global health”
2. The sentence on line 72-73 saying that “the impact of meteorological factors on health…” is a repeat of the first sentence and can be removed. The reference can be added to the other sentence.
3. The sentence structure within the introduction is very basic and reads as a series of short statements, rather than a paragraph that flows from one concept to the next. I would encourage the authors to improve the sentence structure so that it flows nicely.
4. The authors need to combine and condense the sentences starting on line 77 with “high” and ending on line 80 to make the sentence structure flow better. These statements also have no references, which the authors will need to add.
5. Again, in the second paragraph of the introduction, the sentences read as choppy non-flowing statements, and I would encourage the authors to use more complex sentence structure to improve the flow of the paragraph for the reader.
6. I like the last paragraph of the introduction where the authors stated what is missing from research and what their study aims to add.
7. Could the authors clarify about the exclusion criteria – were these diagnoses pre-pregnancy? (line 126)
8. The authors should re-define the abbreviations for the 4 complications in the first sentence on line 139 so that the reader doesn’t have to scroll up to the previous section
9. How did the authors collect the data for the participants? Was there a database used? Did they contact each hospital? I think it would be important to add a line about this
10. Did the authors have any data about previous pregnancy complications for those women who had previous pregnancies? If so, how did they adjust for this, as women who had previous complications are probably at risk for more?
11. Make sure there are references for all the claims about timing of diagnoses during pregnancy. The authors state that GDM is diagnosed between 24-28 weeks and other claims about timing of diagnosis for hypothyroidism, GE and PE so the authors need to have references for all of these claims.
12. Line 255-258: I would ask the authors to please put in p values to help the reader determine the significance of the statements you are making about higher vs lower levels. This will make it easier to follow as the reader is reading the paragraph without jumping separately to the table to see the significance.
13. The last paragraph of the results describing the effects of interactions is very difficult to read. I wonder if the authors could distill this down to a more readable format, maybe with only the most important numbers in the text and the rest in a table for the reader to refer to. It was easy to get lost in all of the numbers and not really understand the point of the paragraph.
14. Line 485 – need a reference for this statement.
15. Line 528: the statement “ interactions between …” is grammatically not correct. I think the authors mean to say “Interactions between extreme meteorological factors…”

---

## Round 0.2 · accepted · Accept

Thank you for carefully addressing the reviewers' comments. One of the reviewers has provided a second review and is appreciative of your attention to detail and thorough addressing of the review comments. Based on this I recommend that the paper be accepted for publication.

Please consider specifying the statistically significant meteorological factors in the abstract as requested by the reviewer.

·

Basic reporting

The comments regarding grammar and clarity throughout the manuscript have been thoroughly addressed. The revised document is clear and unambiguous, meeting professional standards with well-constructed English grammar. The literature review provides sufficient background, and the introduction is coherent and well-structured, supported appropriately by references. The background section effectively outlines the aim of the study and identifies gaps in previous research.

Raw data for tables and figures have been included, with all content appropriately presented and conforming to the standards of a professional article. In the Results section, previous comments have been fully addressed. The section is now coherent, self-contained, and incorporates all relevant findings, adequately responding to the stated hypotheses.

Experimental design

The experimental design of the manuscript clearly articulates the hypothesis and research questions, ensuring they are addressed in a meaningful way and explaining the relevance of the study. Identified gaps are acknowledged, and the statements are clearly explained.
The investigation adheres to academic research standards and is conducted with high technical and ethical rigor.
The methodology is described in detail, providing sufficient information, structure, and content for other researchers to obtain all necessary information and help as reference to conduct similar study.

Validity of the findings

As I have reviewed various research databases, this finding appears to be novel and original, with no concerns regarding duplication. The manuscript’s conclusion is well-articulated and based on the significant findings presented in the main results section, specifically highlighting statistically significant meteorological factors. However, the conclusion in the abstract is overly generalized, stating only that meteorological factors were critical risk factors for pregnancy. The authors should specify the statistically significant meteorological factors in the abstract clearly and precisely, as they have done in the main results section.

Additional comments

no comment